

# A manikin or human simulator—development of a tool for measuring students' perception

Kamil Torres, Phillip Evans, Izabela Mamcarz, Natalia Radczuk and Anna Torres

Chair and Department of Didactics and Medical Simulation, Medical University of Lublin, Lubelskie, Poland

## ABSTRACT

**Background**. Education with the use of medical simulation may involve the use of two modalities: manikins or standardized patients (SPs) to meet specific learning objectives. We have collected students' opinions about the two modalities which can be helpful in planning and evaluating the curriculum process. Although reviews or comparisons of student opinions appear in the literature, it is difficult to find a scale that would be based on a comparison of specific effects that can be obtained in the educational process. In order to fill this gap, an attempt was made to construct a questionnaire.

**Methods**. An experimental version of a questionnaire measuring the final-year students' (273) opinions about the effectiveness of both simulation techniques has been designed on the basis of semi-structured interviews. They were conducted with 14 final-year students excluded from the subsequently analyzed cohort. The scale has been completed, tested and validated.

**Results**. The authors developed a 33-statement questionnaire which contain two scales: teaching medicine with the manikins and with the SPs. Two factors were identified for each scale: Doctor-patient relationship and practical aspects. The scales can be used complementary or separately, as the article reports independent statistics for each scale. The Cronbach's alpha coefficient for the manikin scale is 0.721 and for the SP scale is 0.758.

**Conclusions**. The questionnaire may be applied to medical students to identify their opinions about using manikins and SPs in teaching. It may have an important impact for planning curriculum and implementing particular modalities in accordance with the intended learning objectives.

## INTRODUCTION

Simulation has become a popular tool in undergraduate medical education in many countries (*Ministry of Health, 2012*; *Coffey et al., 2016*; *Alsaad et al., 2017*). The use of medical simulation methods brings several advantages to the process of teaching and learning medicine, *e.g.*, increasing the realism of performed procedures, ensuring a safe environment, minimizing the risks involved in learning how to perform procedures, and enabling students to reproduce specific tasks planned in the scenario, etc. (*Gaba, 2004*;

Corresponding author
Natalia Radczuk,
natalia.radczuk@umlub.pl

*Cleland, Abe & Rethans, 2009*). Commonly used methods like standardized patients or low and high-fidelity manikins can help in achieving these benefits (*Majmudar, Colucci & Landman, 2015a*; *Majmudar, Colucci & Landman, 2015b*; *Bragard et al., 2019*).

Implementation of medical simulation in undergraduate medical education may involve high-fidelity manikins used in simulation scenarios designed for individual courses (*Newcomer, Hatry & Wholey, 2015*). Manikins can help teach students clinical procedures and assess their level of performance. Nowadays, this modality is quite common and useful in achieving numerous educational objectives (*Norman, 2010*).

The study of using manikins in medical education has provided evidence and expertise about the effectiveness of this teaching tool for health care professionals (*Akhu-Zaheya, Gharaibeh & Alostaz, 2013*; *Swamy et al., 2014*). Manikins highlight the importance of experiential learning in introducing and managing various clinical cases without the risk of patient vulnerability and distress (*Ziv, Small & Wolpe, 2000*). Students usually find manikins useful when they practice performing complicated diagnostic procedures or following complex medical algorithms, and the process decreases their stress and increases satisfaction with their clinical achievements (*Cooper, 2004*; *Ahmed et al., 2013*; *Bragard et al., 2019*).

Another teaching technique that has been implemented in medical schools is cooperation with standardized patients. Many authors report employing standardized patients to teach and assess medical students' clinical skills (*Bergin & Fors, 2003*; *Beyea, 2004*; *Nestel & Bearman, 2014*). Sessions with SPs provide broader clinical experience as they allow a well-organized teaching and learning process of selected range of clinical scenarios (*Bokken et al., 2009*; *Cleland, Abe & Rethans, 2009*), development of communication skills, and assessment of professionalism (*Swanwick, 2013*). Students also find the scenarios with SPs more realistic (*Sanko et al., 2012*). When they incorporate various technical and non-technical skills, students become more and more confident in their abilities to provide the best quality care (*Bokken et al., 2009*; *Cleland, Abe & Rethans, 2009*; *Swanwick, 2013*).

Our goal was to develop a tool for measuring students' perception of using two modalities—standardized patients and manikins—in the learning process. The article presents a pilot study to test the psychometric values of the tool. In the process of developing the tool, a semi-structured interview was conducted to gather the knowledge of the phenomenon discussed. Together with the authors' expertise and experience, it was the source of the questionnaire construction. The article presents the process of tool construction as well as the results of factor analysis, reliability and internal consistency.

The construction of a tool to measure students' perspectives on the modalities can be an important contribution to broaden the knowledge in that area that is fundamental to curriculum design and planning. Being aware of students' perspectives, educators can work on building students' knowledge and awareness of learning with manikins and standardized patients, among other things, during training, or during the pre-briefing. Further research in this field could help answer the question, whether increasing students' knowledge regarding the use of manikins and standardized patients translates into the effectiveness of using these modalities in teaching, influences reaching the learning outcomes, or changes students' opinions about these modalities. It is also noteworthy

that no similar tool has been found in the literature that would allow comparison of these two modalities based on the students' perspective. The questionnaires like, for example, the Satisfaction with Simulation Experience Scale (SSES) (*Levett-Jones et al., 2011*) used to measure satisfaction with simulation experience, or the Dundee Ready Education Environment Measure (DREEM) (*Roff et al., 1997*), the Johns Hopkins Learning Environment Scale (JHLES) (*Shochet, Colbert-Getz & Wright, 2015*; *Sengupta, Sharma & Das, 2017*) measuring educational environments, do not include the aspects related to students' opinions about particular instructional methods.

Some authors attempted to compare student opinions and experiences regarding the use of the manikin and SP (*Meerdink & Khan, 2021*; *Liaw et al., 2014*). Reviews on both modalities also appear in the literature (*Cook, Erwin & Triola, 2010*; *Cheng et al., 2015*). However, it is difficult to find works that present tools allowing the comparison of specific effects that can be achieved by working with both techniques. Therefore, we considered to present a tool that would allow to learn students' opinions about the use of the manikin and SP in learning medicine, but in relation to specific skills that can be developed during learning with the use of medical simulation.

Since we did not assume the dimensions that the statements of our method might consist of, we decided to use Exploratory factor analysis (EFA). It is used in academic research and allows summing the items into specific dimensions. EFA is also used in scale development process to condense a large number of items into methodologically sound instrument (*Hooper, 2012*). Exploratory Factor Analysis was conducted separately for the SP and the manikin sections of the questions.

## MATERIALS & METHODS

This study project aimed to develop a tool for measuring students' opinion about using manikins and SPs in learning medicine. Its construction had been preceded by semi-structured interviews with sample students, whose answers constituted the basis for the questionnaire items. It allowed the authors to verify whether their questions arising from the literature review and experience were relevant to the purpose of the study. They referred to the courses integrating clinical skills based on a medical simulation that employed both manikins and standardized patients as teaching modalities. The curricula were designed by an internist with a ten-year clinical experience; a psychologist, who was a specialist in SP-based simulation (completed a course at the University of Cambridge); a surgeon with a twelve-years of clinical experience and an MSc in Clinical Education; and a fellow in infectious diseases with twenty-five years of clinical experience. The teaching faculty had had a four-year-long experience in conducting high-fidelity simulation sessions with manikins and standardized patients.

Ethical approval for the survey was granted by the Medical University of Lublin Bioethics Committee, the Ethical approval no. KE-0254/59/2018, and the survey was performed following the ethical standards laid down in the 1964 Declaration of Helsinki and its later amendments.

## Semi-structured interview

Following the academic principles used for conducting interviews in the 'Handbook of Practical Program Evaluation' (*Newcomer, Hatry & Wholey, 2015*) and building on the literature, the authors of the study prepared a series of open questions. The first stage of the tool development was to conduct semi-structured interviews. The purpose of this stage was to gain information about the use of both modalities in teaching medicine. The responses obtained from the subjects formed the basis for the construction of the questionnaire statements.

An independent psychologist, experienced in the interview techniques, unknown to the students before the study, was commissioned to conduct the interviews as a safeguard against introducing bias. The questions are reported in Appendix 1.

The students' selection for the interviews was made by an open invitation, which was announced on the University's main web page and the web page of the Student Research Association for Medical Simulation at the Medical University of Lublin. The registration was open for two weeks and attracted 14 candidates. The method of registration allowed the participants to remain anonymous to the author. The neutral psychologist made the arrangements for completing the interviews. The time was not limited; therefore, as agreed by the participants, the meetings lasted until the point of saturation of information was reached.

The interviews were conducted in a quiet room, away from the main teaching areas, and considered "a neutral place." At the beginning of each interview, the participants were provided the study description, its aim and methods, and assurance of anonymity. They were allowed to withdraw if they wished. All participating students signed an informed consent form, which gave consent to be involved in the research and be audio-recorded. There was no pressure for time, and the participants could take their time before they gave their answers. They were also allowed to review, modify, or change anything they said during the interview.

The surveyed students' answers formed the foundation for constructing a questionnaire for measuring students' opinions about standardized patients and manikins as simulation techniques.

## The questionnaire-survey
### Characteristics of the study group

The sample population included final-year medical students who had experienced high-fidelity simulation scenarios with a manikin and an SP and practical clinical classes with real patients. 273 out of 383 students in their final year of the medical program completed the survey. The authors of the article did not receive feedback as to why some of the questionnaires were not completed. The surveyed students didn't have extensive experience working in the simulation environment, which was an appropriate condition for conducting a study that included two modalities for such a group of respondents.

The first part of the survey included demographic questions used to collect information about the characteristics of the study group (Table 1). The data were described through

**Table 1  Study participants.**

| | Participants<br>$n = 273$ |
|---|---|
| GENDER<br>n, (%) | |
| Men | 108 (40) |
| Women | 165 (60) |
| AGE | |
| Median | 25.00 |
| Minimum | 23.00 |
| Maximum | 33.00 |
| Lower and upper quartile 25%–75% | 24.00–26.00 |

descriptive statistics, and normality of age distribution was verified with the Shapiro–Wilk's W test.

All data were analyzed using the software STATISTICA Version 12 (StatSoft Inc., Tulsa, OK, USA). The values of the effect size and power of the tests were calculated using G*Power software (*Faul et al., 2007*; *Erdfelder et al., 2009*). The internal consistency of the questionnaires was calculated and expressed as Cronbach's alpha (*Cohen, 1992*). The comparisons were computed with Statistic Calculator (StatPac Inc., Northfield, MN, USA).

The questionnaire was developed based on the relevant literature, the authors' expertise, and the answers coming from the interviewed respondents. Regarding the need to capture students' perceptions about working with the manikin and SP, collaboration was undertaken with two independent psychologists. Their task was to evaluate the phrasing and accuracy of the proposed items, which were formulated based on semi-structured interviews. The criteria according to which statements were qualified for the experimental version of the scale referred to: the level of realism of the statement, the relation of the content of the statement to the program objectives, the relation of the content of the statement to the assumed objectives and outcomes of the course, aspects related to the development of skills and the acquisition of student knowledge and teamwork, the students experience of having to work with a manikin or SP, the opportunity to practice medical procedures, the development of their professionalism, the level of immersion.

In the next step, the research team, taking into account the opinion of the psychologists, decided on the experimental version of the statements. Further qualification of the statements for the final version of the questionnaire was based on the conducted statistics, which are presented below. As a result of the steps taken, such as item content analysis and statistical analyses, the final version of the scale was obtained consisting of 33 statements.

It aimed to provide a detailed analysis of the students' opinions, so the article-based survey was carried out by a psychologist in extensive group sessions in the regular timetable. The purpose of the constructed questionnaire was to compare the perceptions of the students' experience between the manikin and the SP.

The academic principles used for conducting the survey were set out in Greenhalgh's 'Selecting, designing, and developing your questionnaire' (*Boynton & Greenhalgh, 2004*).

The primary version of the questionnaire that was used in the study and then statistically calculated, as shown in Appendix 2, consisted of 46 questions - 23 related to teaching with a manikin and another 23 to an SP. All of the questions were yes/no questions.

In constructing the statements, mirror statements were used to describe both realities—learning based on an SP and on a manikin. This allowed the authors to capture these relationships in a complementary way, to check the consistency of given answers, as well as to choose one part of the tool—concerning an SP or a manikin—depending on the research questions posed.

## RESULTS

Out of 273 students who took part in the survey, 165 (60%) were women, and 108 (40%) were men with a median age of 25 years (range 23 to 33 years) (Table 1).

### Factor analysis for manikin section of questions

The two-component Principal Components Analysis (PCA) was used. The percentage of variance explained by the factors is 28%. A varimax rotation was used to improve the results. The coefficients of the factor-qualified loadings were greater than 0.40.

As a result of the analyses for both the manikin and SP scales, the statements were grouped into factorial sets, which were given the following names based on their meaning:

**Doctor-patient relationship**—assertions related to the development of skills that are relevant to building relationships and doctor-patient mutual understanding.

**Practical aspects**—statements related to the acquisition of practical skills, necessary in the work of a doctor.

The following questions within the manikin section of questions contained the two factors (Table 2).

- Doctor-patient relationship: 3, 10, 11, 12, 13, 14, 15, 18, 21
- Practical aspects: 4, 5, 6, 7, 20

A few questions (number: 1, 2, 8, 9, 16, 17, 19, 22, 23) did not qualify for any factor.

### Factor analysis for SP section of questions

The two-component Principal Components Analysis (PCA) was used. The percentage of variance explained by the factors is 36%. A Varimax Rotation was used to improve the results. The coefficients of the factor-qualified loadings were greater than 0.40.

As a result, the following questions within the manikin questions contained the two factors (Table 3):

- Doctor-patient relationship: 26, 32, 33, 34, 35, 36, 37, 39, 41, 42, 43, 44, 46
- Practical aspects: 24, 25, 27, 28, 29, 30, 31

Again, a few questions (number: 38, 40, 45) did not qualify for any factor.

**Table 2** Factor analysis for the manikin section of questions.

| Question number | Doctor-patient relationship | Practical aspects |
|---|---|---|
| 1 | 0.18 | 0.28 |
| 2 | 0.24 | 0.34 |
| 3 | 0.52 | 0.18 |
| 4 | 0.05 | 0.56 |
| 5 | 0.13 | 0.56 |
| 6 | −0.01 | 0.46 |
| 7 | −0.09 | 0.53 |
| 8 | −0.13 | 0.31 |
| 9 | −0.24 | 0.32 |
| 10 | 0.65 | −0.06 |
| 11 | 0.62 | 0.02 |
| 12 | 0.76 | 0.07 |
| 13 | 0.77 | −0.05 |
| 14 | 0.80 | 0.02 |
| 15 | 0.52 | −0.18 |
| 16 | 0.11 | 0.40 |
| 17 | 0.21 | 0.06 |
| 18 | 0.58 | 0.04 |
| 19 | −0.07 | 0.38 |
| 20 | −0.10 | 0.50 |
| 21 | 0.58 | −0.06 |
| 22 | −0.04 | 0.37 |
| 23 | −0.01 | 0.25 |
| Explained variance | 4.09 | 2.36 |
| Percentage of explained variance | 0.18 | 0.10 |

## Reliability and internal consistency

Cronbach's alpha analysis was conducted to determine the reliability of the scale. The reliability index was calculated for the individual subscales (Doctor-patient relationship and Practical aspects) and for the manikin and SP scales.

The reliability index for the subscales and the scale as a whole on the manikin experience are presented in the Table 4. The Cronbach's alpha coefficient for the manikin scale is 0.721, for the Doctor-patient relationship it's 0.825 and for the Practical aspects subscale it's 0.524.

The reliability index for the subscales and the scale as a whole on the SP experience are presented in the Table 5. The Cronbach's alpha coefficient for the SP scale is 0.758, for the Doctor-patient relationship it's 0.846 and for the Practical aspects it's 0.780.

As a part of scale validation correlations between individual items and overall scale scores were calculated (American Educational Research Association, 2014). The items no. 15, 20, 32, 39, 46, 30, 31 are reversed. As for the Manikin DPR subscale, the results are presented in Table 6.

All of the correlations tested were found to be statistically significant.

**Table 3  Factor analysis for the SP section of questions.**

| Number of question | Doctor-patient relationship | Practical aspects |
|---|---|---|
| 24 | 0.14 | 0.54 |
| 25 | 0.27 | 0.50 |
| 26 | 0.55 | −0.14 |
| 27 | 0.03 | 0.78 |
| 28 | −0.04 | 0.71 |
| 29 | −0.09 | 0.77 |
| 30 | −0.14 | 0.69 |
| 31 | −0.18 | 0.57 |
| 32 | 0.47 | −0.13 |
| 33 | 0.65 | −0.10 |
| 34 | 0.65 | 0.08 |
| 35 | 0.72 | −0.04 |
| 36 | 0.65 | 0.07 |
| 37 | 0.66 | −0.17 |
| 38 | 0.23 | −0.08 |
| 39 | 0.48 | −0.10 |
| 40 | 0.37 | 0.11 |
| 41 | 0.53 | 0.02 |
| 42 | 0.64 | 0.03 |
| 43 | 0.62 | −0.07 |
| 44 | 0.56 | 0.01 |
| 45 | 0.15 | 0.24 |
| 46 | 0.55 | −0.02 |
| Explained variance | 5.02 | 3.24 |
| Percentage of explained variance | 0.22 | 0.14 |

**Table 4  Cronbach's alpha coefficient for the manikin scale.**

| $n = 273$ | The numer of questions | $\alpha$ |
|---|---|---|
| Doctor-patient relationship | 9 | 0.825 |
| Practical aspects | 5 | 0.524 |
| Manikin scale | 14 | 0.721 |

**Table 5  Cronbach's alpha coefficient for the SP scale.**

| $n = 273$ | The numer of questions | $\alpha$ |
|---|---|---|
| Doctor-patient relationship | 13 | 0.846 |
| Practical aspects | 7 | 0.780 |
| SP scale | 20 | 0.758 |

As for the Manikin PA subscale, the results are presented in Table 7.
All of the correlations tested were found to be statistically significant.
As for the SP DPR subscale, the results are presented in Table 8.

**Table 6** Pearson's r correlations between the individual items from the Manikin DPR subscale and the Manikin DPR general score.

| Manikin DPR items n = 273 | | Manikin DPR general score |
|---|---|---|
| No 3 | r | 0.575 |
| | p | <0.0001 |
| No 10 | r | 0.644 |
| | p | <0.0001 |
| No 11 | r | 0.652 |
| | p | <0.0001 |
| No 12 | r | 0.730 |
| | p | <0.0001 |
| No 13 | r | 0.732 |
| | p | <0.0001 |
| No 14 | r | 0.735 |
| | p | <0.0001 |
| No 15 | r | −0.233 |
| | p | <0.0001 |
| No 18 | r | 0.626 |
| | p | <0.0001 |
| No 21 | r | 0.575 |
| | p | <0.0001 |

**Notes.**
r, the Pearson correlation coefficient value; p, significance value of Pearson correlation coefficient.

**Table 7** Pearson's r correlations between the individual items from the Manikin PA subscale and the Manikin PA general score.

| Manikin PA n = 273 | | Manikin PA general score |
|---|---|---|
| No 4 | r | 0.636 |
| | p | <0.0001 |
| No 5 | r | 0.656 |
| | p | <0.0001 |
| No 6 | r | 0.559 |
| | p | <0.0001 |
| No 7 | r | 0.613 |
| | p | <0.0001 |
| No 20 | r | 0.163 |
| | p | <0.007 |

**Notes.**
r, the Pearson correlation coefficient value; p, significance value of Pearson correlation coefficient.

**Table 8  Pearson's r correlations between the individual items from the SP DPR subscale and the SP DPR general score.**

| SP DPR<br>n = 273 | | SP DPR general score |
|---|---|---|
| No 26 | r | 0.554 |
| | p | <0.0001 |
| No 32 | r | −0.183 |
| | p | 0.002 |
| No 33 | r | 0.619 |
| | p | <0.0001 |
| No 34 | r | 0.631 |
| | p | <0.0001 |
| No 35 | r | 0.685 |
| | p | <0.0001 |
| No 36 | r | 0.578 |
| | p | <0.0001 |
| No 37 | r | 0.649 |
| | p | <0.0001 |
| No 39 | r | −0.155 |
| | p | <0.010 |
| No 41 | r | 0.583 |
| | p | <0.0001 |
| No 42 | r | 0.562 |
| | p | <0.0001 |
| No 43 | r | 0.540 |
| | p | <0.0001 |
| No 44 | r | 0.527 |
| | p | <0.0001 |
| No 46 | r | −0.198 |
| | p | <0.001 |

**Notes.**
r, the Pearson correlation coefficient value; p, significance value of Pearson correlation coefficient.

All of the correlations tested were found to be statistically significant.

As for the SP PA subscale, the results are presented in Table 9.

All of the correlations tested were found to be statistically significant, except for the correlation between item no. 31 and general score (GS) ($p = 0.652$). This item has been removed from the final version of the scale.

The statistics conducted allowed us to obtain the final version of the questionnaire, consisting of 33 items, investigating students' perceptions about learning with a manikin and SP. The content analysis of these statements, carried out by the competent judges, confirmed their content and quality consistency with the factors studied.

The available literature as well as the interviews conducted allowed for the construction of a tool that consistently obtains opinions from respondents about the modalities used in medical simulation (manikin and SP). Analyzing the available literature and the previously conducted research, it seemed important to be able to relate students' opinions not only to

**Table 9 Pearson's r correlations between the individual items from the SP PA subscale and the SP PA general score.**

| SP PA<br>n = 273 | | SP PA general score |
| --- | --- | --- |
| No 24 | r | 0.628 |
| | p | <0.0001 |
| No 25 | r | 0.648 |
| | p | <0.0001 |
| No 27 | r | 0.706 |
| | p | <0.0001 |
| No 28 | r | 0.575 |
| | p | <0.0001 |
| No 29 | r | 0.590 |
| | p | <0.0001 |
| No 30 | r | −0.155 |
| | p | <0.011 |
| No 31 | r | −0.027 |
| | p | <0.652 |

**Notes.**

r, the Pearson correlation coefficient value; p, significance value of Pearson correlation coefficient.

their overall impression of working with the manikin or SP, but also to the specific skills that can be acquired by working with these techniques. In response to this perceived gap, we decided to construct an appropriate tool and to allow evaluation using yes/no responses (with regard to each modality separately) rather than through grading or open-ended statements. This makes it seem possible to compare the two modalities and to get a clear overview of the students' perspective.

## DISCUSSION

In this article we have reported development, initial validation and dimensionality of a questionnaire for measuring students' perception about manikins and SPs used for learning in medical simulation. The tool presented in this article captures many aspects related to working with a manikin and an SP, which can be grouped into two areas: related to the development of skills relevant to building doctor-patient rapport and understanding, and to the acquisition of practical skills necessary in the work of the physician. In the *Flössel et al. (2021)* study, the practical aspect of the course implemented using the manikin and SP was taken into account during the survey and was rated highly by the respondents. Other skills, such as communication skills developed during medical simulation classes, were also studied by researchers (*El Naggar & Almaeen, 2020*).

The conducted research allowed the construction of a tool that makes it possible to obtain students' opinion regarding the use of manikins and standardized patients in medical simulation courses. This opinion refers to the specific skills that are assumed to be developed when working with these modalities. Considering the manikin and SP scales separately makes it possible to use them independently to match the research needs. At the

same time, the tool can be the basis for comparing students' opinions on the two modalities. Our main finding is that the questionnaire was worth developing and may subsequently be very helpful in planning curricula and organizing the teaching process making it possible to achieve the assumed educational objectives.

The information obtained from the respondents in the first stage of the study made it possible to construct statements that constitute a scale assessing students' perceptions about working with the two modalities—a manikin and an SP. The need to construct a scale was recognized by the intention to get to know the students' perspective on the use of the manikin and SP in the learning process. This aspect seems to be important when organizing courses using medical simulation, as an additional variable, next to the capabilities and level of technological advancement of the unit, the objectives of the course and the curriculum, the strategy adopted by the teacher to achieve the objectives. All these elements seem to be very important to organize the teaching process in a reasonable and effective way—especially assuming high possibilities of using simulation and the nature of organized classes, that is practical classes. Knowledge of students' perceptions of the modalities is therefore an important input in the process of making decisions on how to effectively implement the curriculum. At the same time, it allows for the selection of the appropriate modality in the context of the possibility of achieving the objectives and incurred costs. Moreover, it seems important to answer the question whether the examined modalities can be treated interchangeably or complementarily, which in educational practice may be of significant importance. The starting point for these conclusions may be to know precisely what are the students' experiences and perceptions of working with a manikin and an SP. Other studies also show that student's experience with medical simulation appears to be an important variable (*Winter-Taylor & Richardson, 2020*; *Badowski, Rossler & Reiland, 2021*).

The work presented herein can be a good starting point for evaluating issues concerning the described phenomena. The developed tool could be used by other medical centers, universities, and educators.

The developed scale can have practical application in the organization of the teaching process. It takes into account two frequently used modalities in medical simulation, *i.e.,* using a manikin and working with a standardized patient, and allows for their comparison. Also, other studies juxtapose these two modalities and provide an opportunity to compare them in the context of their use in medical education (*Bello et al., 2020*; *Meerdink & Khan, 2021*; *Sterz et al., 2022*).

Additionally, it covers two areas—doctor-patient relation and practical aspects—often developed in simulated conditions. Characteristics of students' experiences in terms of these two areas can provide important input into the decision-making process regarding the planning and evaluation of courses implemented using medical simulation.

## Limitations and further directions

The study has its limitations. One of them is the local context. Medical University of Lublin, as other universities in Poland, had to implement medical simulation as a teaching method in undergraduate medical program, however, the directive missed further instructions indicating specific learning outcomes and methods to be included in the curriculum reform.

While we tested particular models of teaching based at our university, other institutions might have different capabilities and approach. Moreover, some of the interfering variables have not been controlled in the survey, as, for example, the frequency of participation in manikin and SP classes and the students' experience with medical simulation during extracurricular activities. Also noteworthy is the fact that according to the factor analysis for the manikin section, the percentage of variance is 28%, while for the SP section is 43%. Hence, if the study is carried out at other universities, it will provide more reliable results. Expanding the research group to post-graduate trainees or health care professionals would also contribute to framing advanced programs of continuing medical education (*Wilbur, Elmubark & Shabana, 2018*).

## ACKNOWLEDGEMENTS

The authors would like to thank the participants for their involvement in this research.

### Funding

The authors received no funding for this work.

### Competing Interests

The authors declare there are no competing interests.

### Author Contributions

- Kamil Torres conceived and designed the experiments, authored or reviewed drafts of the article, and approved the final draft.
- Phillip Evans conceived and designed the experiments, authored or reviewed drafts of the article, and approved the final draft.
- Izabela Mamcarz performed the experiments, analyzed the data, prepared figures and/or tables, authored or reviewed drafts of the article, and approved the final draft.
- Natalia Radczuk performed the experiments, authored or reviewed drafts of the article, and approved the final draft.
- Anna Torres conceived and designed the experiments, analyzed the data, prepared figures and/or tables, authored or reviewed drafts of the article, and approved the final draft.

### Human Ethics

The following information was supplied relating to ethical approvals (i.e., approving body and any reference numbers):

The Medical University in Lublin granted Ethical approval to carry out the study within its facilities (Ethical Application Ref.: KE-0254/59/2018).

### Data Availability

The raw measurements are available in the Supplementary File.

The data is available at Figshare: Mamcarz, Izabela; Torres, Kamil; Evans, Phillip; Radczuk, Natalia; Torres, Anna (2021): A manikin or human simulator –development of a tool for measuring students' perception - DATASET. figshare. Dataset. https://doi.org/10.6084/m9.figshare.15163917.v1.

## Supplemental Information

Supplemental information for this article can be found online at http://dx.doi.org/10.7717/peerj.14214#supplemental-information.

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
