# Peer review of "A manikin or human simulator—development of a tool for measuring students' perception"

_PeerJ, doi:10.7717/peerj.14214_

## Round 0.1 · original submission · Major Revisions

Please pay careful attention to reviewer number one. You may wish to reformulate your manuscript so that you are either reporting on the tool development or the first deployment of the tool. You may indeed have two papers here and separating the tool development from data collection may make for better reporting.

Reviewer 1 ·

Basic reporting

These sections are all fine. I was surprised not to see two publications on this topic included and referenced. Man versus machine: the preferred modality by Sanko, et al.
https://doi.org/10.1111/j.1743-498X.2012.00593.x which was one of the first to study this comparison. Authors may wish to look at this study. It had similar supporting findings.
Structure of the article is good, there are areas where caps are used incorrectly and a look over of the use of consistent terms and spellings of terms is needed - see embedded comments and highlights in PDF.

At times I found this a difficult article to follow - I am not clear on what was compared exactly. Even in table 3 for example I see percentages listed which shows the students opinions about the effectiveness of manikin based and SP based simulation for learning particular clinical objectives, but it was a single group who answered the questions and it was a Likert scale so it is unclear what this is showing.... comparing proportions makes sense if you are comparing 2 or more groups not a single group.....
Table 4 is not clear to me either.

Also in looking at the statistical test selected please address the choice of the Wilcoxon test over chi square or binominal GLM for proportion comparisons.

fix table 6 so the words in the title columns are not cut off weirdly.

is figure 1 showing the number of course that the students encountered using SPs? is figure 2 showing the number of courses that the students encountered using manikins? Figure 3 does not make sense to me. Figure 6 does not make sense.

I feel the best thing out of this article is the tool development. I would like to see the article focus just on this aspect and perhaps a clearer study done using the tool.

Experimental design

There really was not one.
Question was defined, but had to read in between the lines, authors should consider making the question more clear. The aim as written was to establish the opinions of medical undergraduates about their experience of manikins and SPs so that their views are included in a discussion about systematic introduction of simulated patients in undergraduate education. Really there were two aims, development of the tool and the answering of the question posed. the question needs to be made more clear for the reader.
Again I feel the strongest aspect of the paper is the stellar efforts to develop a tool to measure perceptions of students' experiences about two modalities of simulation.

Validity of the findings

the findings are in my opinion inconclusive.
This needs work - it may be all there - but it is difficult to follow and because the question is not well defined I am not sure they answered the question posed in the study. Additionally they need to address the issue brought up about the statistical test selected.

Additional comments

Dear Authors
Thank you for your work in trying to capture the perceptions of learners' experiences with mannequin and SP based simulation. The best part of the study was how you carried out the process to develop the tool. I suggest you publish this separately from the study conducted. Separating the two pieces will add clarity and brevity to the manuscript. Also I think you may need to rethink how you attempted to answer the question or more clearly state your research question(s). Please also consider adding a copy of the tool developed. In addition to some of what I've articulated in this format I have included numerous considerations and suggested edits embedded in the PDF of the manuscript - please se this as well.
Thank you for your time and efforts in conducting this study.

Annotated reviews are not available for download in order to protect the identity of reviewers who chose to remain anonymous.

·

Basic reporting

1. Overall, use of English and structure is of an acceptable standard.

2. Syntax is some areas is not quite right. Specific examples:
Line 13: ‘plan carefully’ —> ‘carefully plan’
Line 16: ‘considered necessary’ —> ‘important’/‘essential’
Line 35: ‘simulation became’ —> ‘simulation has become’
Line 135: ‘a twelve-year clinical experience’ —> ‘twelve-years of clinical experience’
It would be useful to have a native English speaker review (or re-review) before final submission.

3. Some brief sentences are followed by lists of 4-5 (sometimes more) references, this does at times seem imprecise - could some of the references be rationalised. Examples:
Line 40 - 41 - five references used for this sentence. Perhaps break down into multiple smaller sentences with one or two references or instead choose fewer references that incorporate all your points.
Line 43 - 46 - nine references! I think one or two references would evidence your point just as well.

4. You describe cost as being a critical issue in delivering medical simulation (lines 57 - 59) - can you provide evidence of this, or perhaps some evidence of the cost discrepancy between different educational methods?
Similarly line 65 should have a dedicated reference at the end of the sentence (I note appropriate references on line 67).

5. Line 218 should make reference to a figure/table to show the characteristics of participants (I note this is included as Table 2).

6. You refer to 'Cronbach’s Alfa’ a number of times - I believe this should be ‘Cronbach’s alpha'

Experimental design

1. To me this study poses a clear and relevant question, and uses an appropriate mixed qualitative and quantitative methodology to answer this question.

2. Line 145 - 147; use of an independent psychologist to try and reduce bias is a good measure.

3. Line 151 - are there any reasons why students did not complete the survey?

4. Line 155 - 159 - Can you explicitly state whether there were/were not any exclusion criteria, besides being a final year student?

5. Line 196 - 197 There is also evidence about the benefit of a neutral point on the scale, and some suggestion that using a 10 point scale may provide better validity (https://scholarworks.boisestate.edu/cgi/viewcontent.cgi?article=1086&context=ipt_facpubs) Did the authors consider this?

Validity of the findings

1. Authors appropriately identify key limitations regarding generalisability to other student cohorts, as well as that there was no control group with limited experience of one or both types of simulation (manikin or simulated patient).

2. Line 372 - 373 - This seems to be one of your really critical points; use of manikin or simulated patient should depend on what the learning outcomes are. This point is well made and evidenced.

3. Was there any real or perceived risk of bias in that this study was conducted formally within the medical school and by medical school staff? I.e. could there have been any concern from students that they needed to respond in a certain way to this study to avoid perceived repercussion etc. from the medical school.

Additional comments

Thank you for the opportunity to review your work. This was an interesting paper to read that appears well-written and referenced, with some relevant findings. I have acquired a new perspective from this research, that manikins and simulated patients may be better suited to different learning situations; one is not necessarily 'better' than the other - at least from students' perspectives.

---

## Round 0.2 · Major Revisions

Please see the review for your most current version of the manuscript. Should you wish to continue the revision process, I would suggest paying careful attention to this reviewers suggestion, including a more comprehensive expansion of the discussion section. In particular, additional validation must be undertaken for further consideration at the journal.
A second Academic Editor with expertise in this area was consulted in this decision. They have noted that the standards of assessment can be found in Chapter 2 of The Standards for Educational and Psychological Testing (https://www.apa.org/science/programs/testing/standards)

Reviewer 3 ·

Basic reporting

Thank you for submitting the manuscript, which is relatively free of grammar error and easy to read. I believed that after revision, the simulation-related terms in manuscript are acurate. However, in aspect of the structure of the manuscript, I feel that the demographic information mentioned in Lines 191 - 192, as well as Table 1, should be included in the Results session, rather than the Methods session.

Experimental design

It is interesting to see the author teams focused on developing a tool that specifically targetting the measurement of students’ perception and preferences regarding manikin/task-trainers versus human patient simulation (SPs). Although in the field of simulation practice, we tend to have a common sense regarding the applicability of different simulation modalities, few studies have been employed to probe from the perspective of students/learners, which is the ultimate target of simulation instruction/teaching.

As for a study of assessment tool, the authors have conducted literature review, and performed student survey. However. I feel that the there is a need to also perform a expert consensus regarding the proposed items that would to incorporate in the questionaire, regarding the capability, necessity, and the appropriateness of how the questions/items is being approached/phrased, before the actual implementation/distribution of the survey.

Validity of the findings

The current illustration of the Discussion Part seems to be very superficial . The authors should further extend the discussion regarding the need and the impact of this study; how such work could help further facilitate the better use of different simulation modelities in various simulation sessions; comparing the results of this study with other similar studies; as well as the strength and limitation of the study.

---

## Round 0.3 · Minor Revisions

Thank you for resubmitting your manuscript. Writing for publication is of course an iterative process. The manuscript is much improved. Please see the most recent reviews of your manuscript, paying close attention to Reviewer #2. I look forward to your re-submission addressing the most recent reviewers comments and recommendations.

Reviewer 1 ·

Basic reporting

improved with changes

Experimental design

sound

Validity of the findings

good

Additional comments

on line 335 change the term invented scale to either the name of the scale developed or the developed scale.

·

Basic reporting

English language is clear and unambiguous - improved from last draft.

In the abstract, the background does not necessarily make clear why your research is necessary or relevant - please try to explain why there was a need to develop your questionnaire.

I would suggest rationalising/reducing your total number of references if possible. Ideally you can evidence a point with one or two citations rather than four or more at times. See lines 32, 36 + 37, 40 + 41, 75 +76.

Overall the introduction section is probably lengthier than it needs to be. It could be shortened to become more focused and make your points clearer. In essence there needs to be some background as to how and why simulation is used in medical education, and then you need to outline why there is a need to measure students' perceptions of different simulators.
Lines 34 - 46 could be made more brief. Lines 73 - 108 are fairly lengthy - I do not think you need to go into this much detail about the pros/cons of simulation.

You make clear that a similar tool was not found in the literature - you mention this only briefly at the end of the introduction and this could be further elaborated on to explain why your research was needed.

Lines 271 to 277 you report a set of numbers with a comma (e.g. 0,825), I believe the commas should be decimal points.

Experimental design

I note your response to feedback to publish the development of this tool as a standalone paper, and as a result your experimental design appears much clearer.

Your methods appear appropriate and are appropriately referenced.

Lines 234 - 236 you mention exploratory factor analysis - I think this needs to be further explained and/or referenced to make clear what you mean by this and what it involved.

As in section 1 above, you need to make clear why there was a need to develop this research tool - this is only addressed briefly in your manuscript.

Validity of the findings

You appear to have demonstrated that you have designed a validated questionnaire for assessing student perceptions.

Lines 310 - 312 you state that your main finding is that it was worthwhile developing the questionnaire - this needs to be more clearly explained/justified.

You appropriately consider that it would be useful for further study/validation of the tool you have developed, and also recognise the limitations of a single-center study.

---

## Round 0.4 · Minor Revisions

Please make a very clear distinction of what you mean by simulated patients. I think you mean standardized patients (SPs) in the conventional sense - somebody playing the role of an actual patient. The term simulated patient has often come to mean a digitally represented patient. This nuance could be remedied by simply changing the term assuming my assumption is correct.

Also, the manuscript would benefit from another careful copy editing. There are a number of places throughout the manuscript where their continue to be typographical errors and opportunities related consistency and style that would improve readability.

The abstract and introduction are not clear unless one reads the full manuscript. Please make sure these sections are explicitly clear as to what you actually did and found.

---

## Round 0.5 · Minor Revisions

Please address the very minor concerns that the latest reviewer has provided.

Reviewer 3 ·

Basic reporting

I feel that Lines 195-201 better fit to serve as the last paragraph of the Introduction section; and despite that Lines 535-542 has addressed the limitation of the study to certain degree, it was more of a future directions. A separate paragraph to address the limitations of this study is preferred.

Experimental design

no comment

Validity of the findings

no comment

---

## Round 0.6 · accepted · Accept

Thank you for your continued and iterative revisions to your manuscript. Your manuscript has now been recommended for publication.

Reviewer 3 ·

Basic reporting

no comment

Experimental design

no comment

Validity of the findings

no comment